# An Open Database of High-Fidelity, Multi-Reynolds Airfoil Polars for Wind Turbine Blade Design

Francesco Papi<sup>1</sup>, Pier Francesco Melani<sup>1</sup>, Alessandro Bianchini<sup>1</sup>

<sup>1</sup>Department of Industrial Engineering, Università degli Studi di Firenze, Florence, 50139, Italy

5 Correspondence to: Francesco Papi (fr.papi@unifi.it)

Abstract. This paper presents an open-access dataset that has been realized to provide high-fidelity aerodynamic polars for a wide range of wind turbine airfoils, computed using a consistent Computational Fluid Dynamics (CFD) methodology. The database includes lift, drag, and moment coefficients across multiple Reynolds and Mach numbers. Coefficients are computed with both fully turbulent and free transition boundary layers. A blend of the free transition and fully turbulent coefficients is also included. Beyond the stall region, state-of-the-art post-stall extrapolation models are used, the calibration parameters of which are derived from a number of high-fidelity calculations at various angles of attack in separated flow. The data is relevant for wind turbine design, modeling, and simulation, and reflects representative airfoil performance along the span of modern, large-size offshore rotors. The database includes FFA-W3, FFA-W2, FFA-W1, DU, FX-77 and the recently developed OSO families of airfoils. All simulations are performed using validated numerical methods and span a range of conditions typical for utility-scale offshore turbines. The paper discusses the dataset and its key differences with other open-access data, as well as some general trends that can be noted in the aerodynamic coefficients; such a discussion is made possible by the volume of data produced and is specifically tailored to wind turbine applications.

# 1 Background and motivation

Wind turbines face a unique set of operating conditions during their lifetime (Veers et al., 2023). These machines mostly operate inside the Atmospheric Boundary Layer (ABL), often in a highly turbulent flow, and are subject to spatial and temporal variations in wind speed, wind direction, and temperature. Recent large-sized rotor blades may even partially operate between the ABL and the free atmosphere during events of strong atmospheric stability, facing inflow conditions unprecedented to date. Moreover, these machines are constantly exposed to atmospheric elements, while being expected to operate reliably for many years with minimal levels of maintenance. This makes them prone to suffering performance degradation due to soiling and erosion over time.

These challenges are reflected in the evolution of wind turbine blade shapes over the years (Bussel and W, 2022). The core geometry of a wind turbine blade consists of radially-stacked two-dimensional airfoil sections. Early wind turbine designs from the '70 generally adopted pre-World War II standard airfoil shapes designed for aircraft wings from the National Advisory Committee for Aeronautics (NACA) (Abbott et al., 1945). These airfoils mostly featured relatively low thickness-to-chord

40

ratios and narrow optimal operating windows, which made scaling wind turbine blades in an unsteady environment in which wind turbine blades operate challenging. Moreover, they often suffered severe performance degradation due to soiling or small-scale damage, with wind turbine power production dropping nearly 30% in some cases (Airfoils, Where the Turbine Meets the Wind, 2025). These considerations motivated researchers to develop airfoil shapes tailored specifically to wind turbine applications.

Figure 1: Historical evolution of airfoil families for wind turbine design in terms of design balance between aerodynamic efficiency and structural bending stiffness. The performance envelope is computed for each family taking the tip airfoil in clean conditions and the root one in rough conditions. Bendig stiffness is assumed to be proportional to cube of the maximum relative thickness of each airfoil.

Development efforts were significantly accelerated by the development of panel methods, which have been developed since the early 1960s (Erickson, 1990). In particular, the coupling of inviscid panel methods with viscous boundary layer models in the early 1980s made low-speed aerodynamic analyses viable (Drela and Giles, 1987). During this decade the increase in computational power and the development of inverse design approaches, where designers specified the desired pressure distribution and the panel method would compute an airfoil shape (Drela, 1989), significantly catalyzed the development of wind turbine airfoils. A hallmark of this evolution is that, since its beginning, it advanced in parallel within academia and wind turbine manufacturers. However, due to industrial confidentiality, commercial airfoil geometries were generally not disclosed, slowing down the validation of computational methods and the development of shared databases for wind turbine design.

From the start, airfoil designers developed families of airfoils, comprising thicker geometries which privileged structural characteristics and more slender shapes to achieve high aerodynamic efficiency. An overview of the evolution of wind turbine airfoils in terms of aerodynamic efficiency and bending stiffness, which is assumed to be proportional to the third power of the local relative thickness, is provided in Fig. 1. One of the first airfoil families to be publicly issued is the S-series by the

https://doi.org/10.5194/wes-2025-257 Preprint. Discussion started: 24 November 2025 © Author(s) 2025. CC BY 4.0 License.

55

75

National Renewable Energy Laboratory (NREL) in the 1990s (Tangler and Somers, 1995). S-series airfoils feature multiple families with varying thickness-to-chord ratios and have been used in multiple commercial wind turbines (Thresher et al., 1994). However, as shown in Fig. 1, these airfoils are still relatively thin for the structural requirements of modern 100+ meter long blades and are relatively sensitive to roughness changes. In the same period (late 1970s), the FX-77 family of airfoils was developed at the University of Stuttgart (Wortmann, 1978). This family featured airfoils with relative thicknesses between 15.3% and 50%, with the thicker shapes derived from truncating a 34.3%-thick design (flatback airfoil). The University of Delft (Netherlands) has developed several airfoils for wind turbine applications from 1990-onwards (Timmer and Rooij, 2003). They can be easily identified by their name: DU (Delft University) – W (wind) – followed by the year of development and the thickness of the design. Several airfoil families have also been developed by Risø in Denmark, namely the A1 (Bertagnolio et al., 2001; Dahl and Fuglsang, 1998), B1 (Fuglsang et al., 2004) and C2 families (Bak et al., 2008). These airfoils are relatively recent and have been designed with aero-structural considerations in mind (Bak et al., 2014). Risø airfoils, however, are used commercially and are not available open access. The Aeronautical Research Institute of Sweden developed a series of openaccess wind turbine airfoils for wind turbine applications in 1990 (Andres, 1990). They have become very popular among the academic community due to their use in several research reference turbines, such as the DTU 10MW (Bak et al., 2013), IEA 10MW and 3.4MW (Bortolotti et al., 2019), IEA 15MW (IEA Wind, 2020) and IEA 22MW (Zahle et al., 2024). Moreover, they feature high thickness-to-chord ratios and good performance when soiled. Several airfoil families have been designed more recently by various institutions such as ECN (Boorsma et al., 2015; Grasso, 2014), CNER (Méndez et al., 2014) and others (Cheng et al., 2014; Hansen, 2018) using a variety of design and optimization methods. These airfoil families are, however, not open-access, and the shapes of the airfoils are not available publicly. On the other hand, SANDIA has recently developed a family of airfoils specifically tailored for offshore wind turbines (Karcher et al., 2025), i.e., for 100+ meter long blades with high Reynolds (Re) and Mach (Ma) numbers. Indeed, the sheer size of modern wind turbine blades has increased Reynolds numbers dramatically, especially offshore. Very limited experimental validation data exists at these Reynolds numbers due to experimental limitations, which makes the calibration and validation of simulation tools challenging. Despite research showing how these rotating blades feature strong three-dimensional flow characteristics, especially near the blade root (Guntur and Sørensen, 2015; Hand et al., 2001), modelling them as a series of bi-dimensional sections allows for several engineering simplifications, and is the core approach used in most modelling tools. As such, all modern aero-elastic tools are based on polar data and therefore accurate airfoil aerodynamic coefficients are essential to obtain reliable results from these models, which are used in some capacity along the entire design chain of a wind turbine, from initial blade concept development to load verification and certification. In the absence of solid experimental reference data, the aerodynamic coefficients of a given airfoil can vary greatly based on the method used to compute the coefficients and the assumptions made. For instance, the aerodynamic coefficients used in the design of the DTU 10MW, the IEA 15MW, and the IEA 22MW differ, despite all three designs featuring the same FFA-W3 family of airfoils. These uncertainties also make comparisons between different airfoil families difficult. Due to being developed in different times and using different tools, the aerodynamic coefficients used in the design of the aforementioned airfoil families cannot be directly compared in most cases. Moreover,

https://doi.org/10.5194/wes-2025-257

Preprint. Discussion started: 24 November 2025

© Author(s) 2025. CC BY 4.0 License.

100

110

115

even for those families that have been extensively tested experimentally, such as the DU-series of airfoils, the FX-77, or some of the FFA-W3 airfoils, making direct comparisons is challenging due to the coefficients being computed by different researchers in different wind tunnels, with different surface finishes, inflow turbulence, blockage ratios, and Reynolds and Mach numbers. In fact, the complex physics at play cause the performance of an aerodynamic shape to be strongly influenced by minor variations in external shape or surface finish, and changes in inflow conditions.

In addition, significant uncertainty is also present in the post-stall region. In fact, while aerodynamic coefficients where the flow is attached are typically computed using either viscous panel methods or computational fluid dynamics (CFD) or are derived from experimental measurements at moderate angles of attack, post-stall extrapolation methods are ubiquitously used to empirically extend the coefficients to a -180° to +180° range. Once again, depending on the method and the assumptions used, estimations can vary greatly. For instance, the maximum drag value within the coefficients used in the NREL 5MW RWT designs in approximately 1.3 before stall-delay correction, while this value is 1.3 to 1.5 in the ones of the DTU 10MW and IEA 15MW and IEA 22MW, depending on the airfoil thickness. Higher values are instead found experimentally, such as in the work of (Timmer, 2020), which measured maximum drag coefficients above 1.8 for a selection of wind turbine airfoils. Similar values are also recommended in some of the original post-extrapolation models, like that of (Viterna and Janetzke, 1982) that suggest a value of approximately 1.8 depending on the specific characteristics of the airfoil such as camber, relative thickness and leading-edge radius. These differences cannot be imputed to Reynolds number alone (Battisti et al., 2020) and indicate significant modelling uncertainty in this area, which is important for example for ultimate blade load estimation in parked conditions.

Due to the issues described so far, computing reliable airfoil coefficients can be challenging for wind energy practitioners who wish to build their own blade design, as it requires significant expertise. It is, in fact, not uncommon to see coefficients computed using low-order viscous panel methods being used in recent research efforts (Gupta et al., 2024), introducing significant uncertainty in drag coefficient estimation and in the near stall region (Ramanujam et al., 2016).

This work aims to address these issues by computing a dataset of high-Reynolds airfoil coefficients that can serve as a common research basis for modern wind turbine design. One of the key aspects of this effort is the fact that the airfoil polars have been computed using the same modelling choices and inflow conditions for the various families included in the dataset, thus ensuring comparability. Moreover, the coefficients included in this dataset take full advantage of CFD and are computed including compressibility, the effects of which are becoming increasingly relevant for modern wind turbines (Vitulano et al., 2025). Multiple airfoil families are included in the dataset, allowing for fair comparisons between airfoil families. Coefficients are

Multiple airfoil families are included in the dataset, allowing for fair comparisons between airfoil families. Coefficients are computed using consistent state-of-the-art numerical methods, which are described in the following sections of this study.

#### 2 Wind turbine airfoils

The dataset analyzed herein contains airfoil coefficients with varying Reynolds and Mach numbers of several open-access commonly used wind turbine airfoils. The dataset includes the FFA-W1, FFA-W2, FFA-W3, DU, FX77 and OSO airfoil

families. One notable exception amongst open-access airfoils is represented by NREL S-series airfoils (Tangler and Somers, 120 1995), as they are relatively slender for modern blade designs. In fact, only airfoils with thickness to chord ratios greater than 21% are included in the dataset. The full list of geometries is shown in Fig. 2 and is summarized in Table 1.

**Table 1:** Airfoils included in the dataset. Free transition: simulations with boundary layer transition (clean blade). Fully turbulent: simulations without boundary layer transition (soiled blade). Blend refers to a 70% free transition and 30% fully turbulent interpolation, to be used as reference for blade design.

| Airfoil      | relative fully turbulent |          | free transition | blend        | Pressure<br>Coefficient | Skin<br>Friction |  |
|--------------|--------------------------|----------|-----------------|--------------|-------------------------|------------------|--|
| FFA-W3-211   | 21%                      | <b>√</b> | ✓               | <b>√</b>     | <b>√</b>                | <b>√</b>         |  |
| FFA-W3-241   | 24.1%                    | ✓        | ✓               | ✓            | $\checkmark$            | $\checkmark$     |  |
| FFA-W3-270   | 27%                      | ✓        | ✓               | $\checkmark$ | $\checkmark$            | $\checkmark$     |  |
| FFA-W3-300   | 30%                      | ✓        | ✓               | ✓            | $\checkmark$            | $\checkmark$     |  |
| FFA-W3-332   | 33.2%                    | ✓        | ✓               | $\checkmark$ | $\checkmark$            | ✓                |  |
| FFA-W3-360   | 36%                      | ✓        | ✓               | $\checkmark$ | $\checkmark$            | $\checkmark$     |  |
| DU08-W-210   | 21%                      | ✓        | ✓               | $\checkmark$ | $\checkmark$            | $\checkmark$     |  |
| DU00-W-212   | 21.2%                    | ✓        | ✓               | $\checkmark$ | $\checkmark$            | $\checkmark$     |  |
| DU00-W2-250  | 25%                      | ✓        | ✓               | $\checkmark$ | $\checkmark$            | ✓                |  |
| DU97-W-300   | 30%                      | ✓        | ✓               | $\checkmark$ | $\checkmark$            | $\checkmark$     |  |
| OSO-21-WT2   | 21%                      | ✓        | ✓               | $\checkmark$ | $\checkmark$            | ✓                |  |
| OSO-24-WT2   | 24%                      | ✓        | ✓               | $\checkmark$ | $\checkmark$            | ✓                |  |
| OSO-27-WT2   | 27&                      | ✓        | ✓               | $\checkmark$ | $\checkmark$            | ✓                |  |
| OSO-30-WT2   | 30%                      | ✓        | ✓               | $\checkmark$ | $\checkmark$            | ✓                |  |
| FX77-W-258   | 25.8%                    | ✓        |                 |              | $\checkmark$            | ✓                |  |
| FX77-W-343   | 34.3%                    | ✓        |                 |              | $\checkmark$            | ✓                |  |
| FFA-W2-210   | 21%                      | ✓        |                 |              | $\checkmark$            | ✓                |  |
| FFA-W1-211   | 21.1%                    | ✓        |                 |              | $\checkmark$            | ✓                |  |
| FFA-W1-242   | 24.2%                    | ✓        |                 |              | $\checkmark$            | ✓                |  |
| FFA-W1-271   | 27.1%                    | ✓        |                 |              | $\checkmark$            | ✓                |  |
| NACA63(3)418 | 18%                      | ✓        |                 |              | $\checkmark$            | $\checkmark$     |  |

145

Figure 2: Airfoils included in the present database grouped by airfoil family.

Modern wind turbine blades, particularly those of offshore machines, are increasingly large. As such, chord-based Reynolds number can exceed 15\*10<sup>6</sup> (Karcher et al., 2025). Reynolds (Eq. 1) and Mach (Eq. 2) number are defined as:

$$Re = \frac{\rho c U}{\mu} \tag{1}$$

$$Ma = \frac{U}{\sqrt{\gamma RT}} \tag{2}$$

where  $\rho$  is the air density,  $\mu$  is the dynamic viscosity of the fluid,  $\gamma$  is the ratio of heat capacity at constant pressure and at constant volume, R is the gas constant and T is the temperature of the fluid. The mean values assumed in this study and their measurement units are summarized in Table 2, which also reports the reference conditions in terms of pressure P and turbulence intensity ti considered for the simulations. A turbulence intensity of 0.35% is considered to match the conditions where the computational model is tuned and validated, as explained in detail in section 3.1.

All the simulations performed in this study model the fluid as compressible. This is a key distinction if compared to the aerodynamic coefficients used in common academic reference wind turbines (Bortolotti et al., 2019; IEA Wind, 2020; Zahle et al., 2024), where the flow is modelled as incompressible. This simplification is increasingly limiting, as recent studies have shown how, due to high tip-speeds and local flow acceleration, transonic flow is possible in certain conditions near the blade tip (De Tavernier and Terzi, 2022), a condition which cannot be appropriately modelled if the flow is considered incompressible. On the other hand, (Vitulano et al., 2025) have shown that URANS simulations are a viable way of predicting airfoil performance in transonic conditions. In addition, compressibility is known to have an effect on the aerodynamic

coefficients, particularly on the slope of the lift coefficient in the pre-stall region, which tends to increase as the Mach number increases (Gudmundsson, 2014).

Table 2: Parameters user in the current study.

| Parameter | value                      |  |  |
|-----------|----------------------------|--|--|
| ρ         | $1.225 \text{ kg/m}^3$     |  |  |
| $\mu$     | 1.77*10 <sup>-5</sup> Pa s |  |  |
| γ         | 1.4                        |  |  |
| R         | $287.05\ s^2/m^2K^{1/2}$   |  |  |
| T         | 273.15 K                   |  |  |
| P         | 101325 Pa                  |  |  |
| ti        | 0.35%                      |  |  |

The second key improvement of this dataset is the fact that multiple Reynolds numbers are considered. In fact, the Reynolds number can vary greatly during operation in actual wind turbines, as the relative inflow velocity changes due to the varying wind speed and rotor speed. In order to select the most appropriate combination of Reynolds and Mach numbers for the various airfoils, the IEA 22MW blade design was assumed as a reference, as it is the most recent and representative of the size and characteristics of upcoming offshore rotors. Individual airfoils are assigned to a representative span of the blade based on their thickness, as thicker airfoils are generally used in the inner part of the blade, while thinner ones are preferred in the outboard part to improve aerodynamic efficiency. Once the average chord of the blade section is computed, the inflow velocity for a given Reynolds number (Eq. 3) can be computed by inverting Eq. 1:

$$U = \frac{Re\mu}{\rho c} \tag{3}$$

The chord length, Reynolds number, inflow velocity, and Mach number ranges assumed for each airfoil relative thickness are shown in Table 3. Five Reynolds numbers are simulated for each airfoil: 2.5\*10<sup>6</sup>, 5\*10<sup>6</sup>, 10\*10<sup>6</sup>, 15\*10<sup>6</sup>, 20\*10<sup>6</sup>. The portion of blade span upon which the airfoils are assumed to insist are shown in Fig. 3.

Table 3: Inflow conditions on the various airfoils based on relative thickness.

| span (%)  | thickness (%) | chord (m) | U (m/s)      | Re                   | Ma          |
|-----------|---------------|-----------|--------------|----------------------|-------------|
| > 85%     | 

180

185

| 10% - 25% | > 38% | 7.147 | 5.1 - 40.4 | $2.5*10^6 - 20*10^6$ | 0.01 - 0.12 |
|-----------|-------|-------|------------|----------------------|-------------|

Figure 3: Assumed range of relative thickness as a function of blade span on the IEA 22MW blade.

#### 3 Methods

All data included in this dataset is computed using two-dimensional CFD. More specifically, compressible Unsteady Reynolds170 Averaged Navier-Stokes (URANS) simulations are performed in Ansys Fluent 23R2. The adoption of the URANS approach
is not necessarily intended to capture unsteady flow features, rather time-dependent simulations are preferred over a steadystate approach as the former have shown to be less prone to numerical instabilities and more accurate especially when laminar
to turbulent boundary layer transition is modelled. The numerical set-up is based on a previously validated approach (Balduzzi
et al., 2021; Papi et al., 2021; Venturi et al., 2024). The working fluid is dry air at sea-level pressure and 0 °C temperature
(Table 2) and is modelled as an ideal gas.

An open-field, bullet-shaped domain is used as shown in Fig. 4. The domain boundaries are placed 600-chords away to avoid interference with the pressure distribution on the airfoil, as discussed in more detail in Section 3.1. The angle of attack is changed by rotating the airfoil inside the domain via a sliding interface. While not presently included in this dataset, this approach allows also for dynamic simulations to be performed. The  $k - \omega$  SST turbulence closure (Menter, 1994) is used in fully turbulent simulations. The numerical decay of turbulence from the inlet to the airfoil is prevented using ad-hoc source terms in the k and  $\omega$  transport equations: this avoids artificially large length-scales and consequent contamination of the solution in non-turbulent regions due to excessive eddy viscosity (Spalart and Rumsey, 2007). Boundary layer transition in the free transition simulations is modelled using the four-equation  $\gamma - Re_{\theta}$  transition model (Langtry and Menter, 2009; Menter et al., 2004). In order to fine-tune the characteristics of the transition model, the correlation developed by Drela (Drela, 1998) for the transition critical Reynolds number is used, as explained in more detail in Section 3.2.

After initial steady RANS initialization, the simulations are run for 50 chord-based throughflow times or until numerical convergence, where numerical convergence is achieved if the percentage variation over the first and last values used to compute the mean aerodynamic coefficients is less than 0.5%.

Figure 4: Domain used in computations.

## 3.1 Numerical model tuning and validation

Two-dimensional CFD computations of external aerodynamic bodies can be particularly sensitive to boundary conditions. To ensure truly open-field conditions, i.e., no artificial acceleration or distortion of the freestream, the boundaries need to be placed extremely distant from the airfoil (Golmirzaee and Wood, 2024). As highlighted by (Sorensen et al., 2016), a smaller domain can be used if a point-vortex correction for inlet velocity is considered. While such a correction was not used in this study, it highlights the influence of lift on the flow-field distortion, thus large domains are required particularly in high-lift situations. The influence of domain size on the predicted performance in fully turbulent conditions at a Reynolds number of 15\*10<sup>6</sup> is shown for the DU00-W-212 airfoil as a representative example in Fig. 5. From a perusal of the figure, it is apparent how boundary distance particularly influences the estimation of the drag coefficient, and it is important to correctly capture peak lift-to-drag ratio. In particular, as already suggested during the comparison rounds of the AVATAR project (Ozlem et al., 2017), the thresholds usually found in most of the literature studies (around 30 chords (Golmirzaee and Wood, 2024)) are apparently not sufficient to ensure robust results. In the present study, the value of 600c (reported in solid lines in the graphs) has been adopted.

215

220

Figure 5: Lift (a), Drag (b) and Lift to Drag (c) ratio as a function of the angle of attack (a, c) and the lift coefficient (b) with various domain sizes.

Focusing on the grid structure, a triangular unstructured grid combined with a prismatic boundary layer mesh is used in this study. The prismatic boundary layer mesh features 85 layers moving away from the airfoil's surface. This guarantees that the boundary layer is fully contained inside the prismatic layer in all tested conditions. The height of the first cell on the blade surface ensures a y<sup>+</sup> value below 1 for all cases. The final mesh used for the simulations of the DU00-W-212 airfoil used in the validation of the numerical approach is shown in Fig. 6. Insensitivity to the mesh was ensured by comparing results obtained in free transition conditions at a Reynolds number of 15\*10<sup>6</sup> of the DU00-W-212 airfoil (Fig. 7). Free transition computations are considered more demanding from this point of view, as correctly capturing the transition point can require finer chordwise resolution than a fully turbulent computation (Menter et al., 2021). The free-stream mesh in the vicinity of the airfoil is parametrized depending on the number of elements along the airfoil itself, therefore increasing the number of elements along the airfoil also increases the total number of mesh elements in the domain, as shown in Table 4. To obtain grid-insensitive results, especially between 5° and 12° of angle of attack, the M3 grid was used in the subsequent simulations.

Figure 6: Triangular mesh used in computations. (a) wake refinement and rotating region around the blade, (b) blade refinement (c) boundary layer mesh.

Table 4: Total and chordwise number of elements for tested computational grids.

| <br>mesh | chordwise elements | total elements |
|----------|--------------------|----------------|
| M1       | 600                | 419128         |
| M2       | 1000               | 543924         |
| M3       | 1600               | 748674         |
| M4       | 2600               | 1298248        |

Figure 7: Lift (a), Drag (b) and Lift to Drag (c) ratio as a function of the angle of attack with varying mesh sizes. Reynolds number of 15\*106, turbulence intensity 0.34%, reference data from (Ozlem et al., 2017).

Figure 8: Lift (a), Drag (b) and Lift to Drag (c) ratio as a function of the angle of attack with varying timestep sizes. Reynolds number of 15\*106, turbulence intensity 0.34%, reference data from (Ozlem et al., 2017).

A sensitivity to the timestep length is shown in Fig. 8. While lift coefficient is mostly insensitive to the chosen timestep, drag coefficient decreases if the timestep is decreased from 5 timesteps per chord through-flow time to 10 timesteps for chord through-flow time. Further decreases do not lead to additional improvement. Based on this analysis, 20 timesteps for chord through flow times are used in the computation of the database.

## 3.2 Quantifying the effects of compressibility

The numerical set-up was verified with respect to the numerical simulations of the DU00-W-212 airfoil performed by (Sorensen et al., 2016). The lift to drag ratio for the DU00W-212 airfoil at a Reynolds number of 3\*10<sup>6</sup> and a Mach number of 0.075 is shown in Figure 9 (a). The pressure and skin friction coefficients at an angle of attack of 4° obtained through incompressible fully turbulent simulation of the DU00-W-212 in the same flow conditions are shown in Figure 9 (c, d). The data is overlayed to the numerical simulations from (Sorensen et al., 2016), where the flow is considered incompressible in all the simulations. In addition to highlighting very good agreement between the current set-up and the reference data, Figure 9 (a) also shows a 2.2% decrease in Lift to Drag ratio at the angle of attack of peak efficiency of 8°. This measurable reduction in efficiency is present despite the relatively low inflow Mach number of 0.075.

Figure 9: Lift (a), Drag (b) and Lift to Drag (c) ratio as a function of the angle of attack (a, c) and the lift coefficient (b) with varying mesh sizes. Reynolds number of 15\*10<sup>6</sup>, turbulence intensity 0.34%, Mach number of 0.075. Reference data in red from (Sorensen et al., 2016) shows the spread in model prediction that was found during the AVATAR project.

# 3.3 Boundary layer transition modelling

Laminar-to-turbulent transition is modeled with the  $\gamma - Re_{\theta}$  formulation, which uses two additional transport equations, one for the intermittency  $\gamma$  and one for the momentum thickness Reynolds number  $Re_{\theta}$ . The model relies on empirical correlations for the critical Reynolds number  $(Re_{\theta,t})$  and transition length, which are used to trigger turbulence production in the laminar boundary layer. Thus, they directly influence the position of the transition point and extension of the transition region. Despite being capable of modelling all kinds of boundary layer transition, the original correlations (Langtry and Menter, 2009) were

https://doi.org/10.5194/wes-2025-257 Preprint. Discussion started: 24 November 2025 © Author(s) 2025. CC BY 4.0 License.

developed with turbomachinery flows in mind and are optimized for *bypass transition* occurring at freestream turbulence values higher than 1%. Despite working inside the ABL however, modern wind turbine airfoils operate at notably lower turbulence levels, as the relatively steady tangential velocity due to the turbine's rotation is the main contributor to the local inflow, rather than the unsteady incoming wind. Such low free-stream turbulence inflows are dominated by the natural transition mechanism (Lobo et al., 2023). For this reason, in this work the correlation developed by (Drela, 1998) for the critical Reynolds number  $Re_{\theta t}$  is used (Eq. 4):

$$Re_{\theta t}(H, N_{cr}) = 155 + 89 \left[ 0.25 * \tanh\left(\frac{10}{H - 1} - 5.5\right) + 1 \right] N_{cr}^{1.25}$$
 (4)

The correlation depends on the amplification factor  $N_{cr}$  and the boundary layer shape factor H. The former can be modelled as a function of the free-stream turbulence intensity (Eq. 5):

$$N_{cr} = f(ti) = -8.43 - 2.4 \ln \left( \frac{2.1 \tanh(\frac{ti}{2.7})}{100} \right)$$
 (5)

or used as a calibration factor, as done in this study. The boundary layer shape factor, i.e., the ratio between boundary layer displacement thickness and momentum thickness, cannot be computed directly, as the  $\gamma - Re_{\theta}$  transition model is based on local variables only, and must be approximated with an empirical correlation. In this work, the correlation developed by White (White and Madani, 2006), based on the experiments of Thwaites (Thwaites, 1949) on flat plates under streamwise pressure gradient was used (Eq. 6):

$$H(\lambda_{\theta}) = 2 + 4.14x - 83.5x^{2} + 854x^{3} - 3337x^{4} + 4576x^{5}$$

$$x = 0.25 - \lambda_{\theta}$$
(6)

where  $\lambda_{\theta}$  is the local pressure gradient, computed locally from the resolved flow field. Once again, the approach was validated with respect to high-Reynolds experimental tests from the AVATAR project (Pires et al., 2016). A comparison between the experimental results and numerical ones at a Reynolds number of  $15*10^6$  is shown in Fig. 10. Although the dataset contains measurements for various Reynolds numbers, ranging from  $3*10^6$  to  $15*10^6$ , the tests differ not only in Reynolds number but also in turbulence intensity. Therefore, the test at Re  $15*10^6$  was chosen for calibration of the numerical models as this condition is the most representative for very large offshore blades. As shown in Fig. 10, while the default correlation was found to produce good results in the conditions tested herein, using the correlation developed by Drela (eq. 3) with  $N_{cr} = 6$  reduced the discrepancy in lift-to-drag ratio at low angles of attack, and was thus chosen for the computation the entire dataset. The free-stream turbulence intensity (Table 2) of the simulations was set to 0.35%, which matches the AVATAR experiments at Re  $15*10^6$ .

Figure 10: Lift (a), Drag (b) and Lift to Drag (c) ratio as a function of the angle of attack (a, c) and the lift coefficient (b). Experiments from the AVATAR project (Pires et al., 2016) are compared to the results obtained with the numerical set-up described in section 2 and various correlations for the  $\gamma - Re_{\theta}$  transition model on the DU00-W-212 airfoil. Reynolds number of 15\*10<sup>6</sup>, turbulence intensity 0.34%, reference data from (Ozlem et al., 2017).

## 3.4 Post-stall extrapolation

As per common practice, the raw lift and drag coefficients were first manually cleaned by removing inaccurate/unconverged results, generally close or after the positive or negative stall points. Clean coefficients were then extrapolated to the full +/-180° angle of attack range using empirical post-stall correlations. Differently from other studies, special attention was given to the extrapolation process. More in detail, choice and calibration of the post-stall extrapolation method was driven by dedicated scale-resolving Stress Blended Eddy Simulations (SBES) simulations, which were carried out for the FFA-W3-241 airfoil at a Reynolds number of 10\*10<sup>6</sup> and selected angles of attack (see Fig. 11).

SBES blends a LES turbulence closure in the free stream and wake of the airfoil when the grid resolution allows it, and a URANS approach in the boundary layer region (Menter, 2021). Simulations are performed in Ansys Fluent 24R1 GPU-accelerated solver using the  $k-\omega$  SST turbulence closure in the airfoil boundary layer and the WALE LES sub-grid scale model in the free-stream. The airfoil section is extruded six chords to allow for spanwise flow non uniformities to fully-develop and avoid the "lock-in" of wake vortex structures and the consequent overestimation of force fluctuations (Mittal and Balachandar, 1995).

Figure 11: Q-criterion isosurfaces for the FFA-W3-241 airfoil at 30° (a, c) and 90° (b, d) angle of attack from scale-resolving SBES simulations. Reynolds number of 10\*10<sup>6</sup>. Side view (a, b) and top view (c, d).

This set-up is compliant with the recommendations found in (Sørensen and Timmer, 2017), where at least four chords of spanwise distance are recommended for high angle of attack simulations. The boundary layer is modeled with a series of prismatic elements, ensuring y+ values at the airfoil surface below 1. The mesh in the immediate vicinity of the airfoil is also prismatic, while a polyhedral mesh is used in the airfoil wake, and is chosen due to generally lower numerical diffusion and improved gradient reproduction with respect to tetrahedral cells (Wang et al., 2021). Four-hundred elements are used along the airfoil in both the chordwise and spanwise directions. This choice is consistent with the state-of-the-art of scale resolving simulations (Manolesos and Papadakis, 2021; Sørensen and Timmer, 2017) is the chordwise direction, while the total number of elements in the spanwise direction is limited due to solver memory constraints. Due to the expected occurrence of large-scale, strongly three-dimensional flow structures in the deep-stall cases analyzed herein, priority is given to the total resolved blade span rather than to the spanwise discretization. Nevertheless, an aspect ratio of approximately 1.5-1.8 is achieved for the

cells in the airfoil's proximity, quickly transitioning to lower values in the polyhedral wake region. The full domain totals approximately 24\*10<sup>6</sup> elements. Similarly to (Sørensen and Timmer, 2017) the domain extends fifty chords from the airfoil and is bullet-shaped (see Fig. 4). Simulations are first initialized with a RANS simulation and then run for 300 chord-based through flow times:

$$\tau = t * U/c \tag{7}$$

or until numerical convergence of the mean lift and drag coefficients. Results are considered converged when the mean values of the aerodynamic coefficients vary less than 2% over the final 70  $\tau$ . Lift and drag coefficients are then averaged starting from  $\tau = 60$ . The strongly three-dimensional characteristics of the flow at both 30° and 90° of flow incidence especially can be seen in Fig. 11. The total computational cost for SBES simulations is approximately 1400 GPU-hours using a single NVidia A100 80GB GPU.

After initial testing of various post-stall extrapolation models such as those developed by (Viterna and Janetzke, 1982), (Spera, 2008), (Beans and Jakubowski, 1983) and (Kirke, 1998), the choice was narrowed down to the models developed by (Montgomerie, 2004) and more recently by (Battisti et al., 2020) based on their agreement with scale resolving simulations (Fig. 12). Based on the comparison of the two methods with SBES simulations of the FFA-W3-241 airfoil (Fig. 12), drag is extrapolated to +/- 180° using the method by (Battisti et al., 2020), while the Montgomerie method (Montgomerie, 2004) is used for lift. Moment coefficient is extrapolated using the relation in (Laino and Hansen, 2002).

Figure 12: Lift (a) and Drag (b) coefficient for the FFA-W3-241 airfoil at a Reynolds number of 10\*10<sup>6</sup>. Two-dimensional CFD simulations extrapolated with the method proposed by (Montgomerie, 2004) and (Battisti et al., 2020) compared to scale-resolving SBES simulations.

After post-stall extrapolation, fully turbulent and free transition coefficients are blended with a 70% free transition and 30% fully turbulent ratio. These sets of coefficients can be considered representative of a new, clean blade, while also including the

effect of aerodynamic deterioration which may occur during operation. This practice is fairly common in academia and has been followed in the design of the latest generation offshore reference rotor, the IEA 22MW RWT (Zahle et al., 2024).

### 330 3.5 Static parameters

In a final step, the static parameters used in the Beddoes-Leishman (BL) type dynamic stall models underlying most aeroelastic tools (Leishman, 2016), are computed from the obtained polar data. A detailed description of each parameter can be found in Table 5. Although not strictly related to the airfoil dynamic behavior, these coefficients represent two-thirds of the parameters necessary to calibrate the model and have a strong influence on the overall accuracy (Melani et al., 2024). The resulting coefficients are provided in the database in AeroDyn format (Jonkman et al., 2025).

Table 5: Static parameters for Beddoes-Leishman type dynamic stall models used in the current study.

| Symbol                                                      | Description                                                           | Units       | Fitting equation                                                                                                                                                                               | Source        |
|-------------------------------------------------------------|-----------------------------------------------------------------------|-------------|------------------------------------------------------------------------------------------------------------------------------------------------------------------------------------------------|---------------|
| $C_{N\alpha}$                                               | $C_n$ curve slope in the linear region                                | rad-1       |                                                                                                                                                                                                | $C_n$         |
| $\alpha_0$                                                  | angle of attack at $C_n = 0$                                          | rad         |                                                                                                                                                                                                | $C_n$         |
| $C_{D0}$                                                    | drag coefficient at $C_n = 0$                                         | -           |                                                                                                                                                                                                | $C_d$         |
| $C_{M0}$                                                    | moment coefficient at $C_n = 0$                                       | -           |                                                                                                                                                                                                | $C_m$         |
| $\eta_e$                                                    | $C_c$ recovery factor                                                 | -           | $\eta_e C_c^{pot} = \eta_e C_{N\alpha} (\alpha - \alpha_0) \tan \alpha$                                                                                                                        | $C_c$         |
| $\alpha_1$ $S_1$ $S_2$                                      | parameters for Leishman exponential fitting of $f$                    | rad         | $f = \begin{cases} 1.0 - 0.30e^{\left(\frac{\alpha - \alpha_1}{S_1}\right)} & \alpha \le \alpha_1\\ 0.04 + 0.66e^{\left(\frac{\alpha_1 - \alpha}{S_2}\right)} & \alpha > \alpha_1 \end{cases}$ | $C_n$         |
| k <sub>0</sub> k <sub>1</sub> k <sub>2</sub> k <sub>3</sub> | parameters for Beddoes fitting of $x_{CP}$                            | -<br>-<br>- | $x_{CP}(f) = k_0 + k_1(1 - f) + k_2 \sin(\pi f^{k_3})$                                                                                                                                         | $C_n$ , $C_m$ |
| $C_{N1}$ $D_{CC}$ $\widehat{k}_1$                           | critical normal load parameters for the fitting of static $C_c$ curve | -           | $C_c = \hat{k}_1 + C_c^{pot} \sqrt{f} f^{D_{CC}( C_n  - C_{N_1})}$                                                                                                                             | $C_n, C_c$    |

The parameter extraction follows the workflow outlined in (Melani et al., 2024). For the sake of brevity, the process is described only for positive angles of attack, but the same considerations apply to the negative ones. As a preliminary step, lift  $(C_l)$  and drag  $(C_d)$  coefficients are converted to the normal  $(C_n)$  and chordwise  $(C_c)$  ones used in the BL formulation. Parameters governing the airfoil behavior in the *attached* and *separated* flow regimes are then determined sequentially.

Figure 13: Visual representation of the computation of the different parameters required by AeroDyn dynamic stall models from static polar data: (a-c) attached flow coefficients (d-f) separated flow coefficients.

For the attached flow regime, constants  $\alpha_0$ ,  $C_{N\alpha}$ ,  $C_{D0}$ ,  $C_{M0}$ ,  $\eta$  are directly obtained from the  $C_n$ ,  $C_c$ , and  $C_m$  curves. The zero-lift angle of attack  $\alpha_0$  is located at the intersection of the  $C_n(\alpha)$  curve with the  $\alpha$  axis, while  $C_{M,0}$  and  $C_{D,0}$  are the moment and drag values at  $\alpha = \alpha_0$  (see Fig. 13(a-c)). The curve slope  $C_{N\alpha}$  is evaluated via linear regression of the data in the linear region, delimited in Fig. 13 (a) by the interval  $[\alpha_{lin,neg}, \alpha_{lin,pos}]$ . This accounts for possible noise in the static data. In this work, an innovative automated method to identify this region is adopted: i) progressively narrower intervals, centered on  $\alpha_0$ , are considered; ii) for each interval, data is fitted via linear regression, using the corresponding  $R^2$  as a linearity index. iii) the linear region is identified as the region where  $R^2 > 0.99$  at both bounds. Finally, the recovery factor  $\eta_e$  is selected by minimizing the discrepancy between  $\eta_e C_c^{pot}$  (see Table 5) and the pressure component of chordwise force  $C_c^{stat} = C_l \sin \alpha - (C_d - C_{D,0}) \cos \alpha$  in the linear region (see Fig. 13 (b)).

In the BL model, flow separation is described by the static separation point (f) and center of pressure  $(x_{CP})$ . The separation point is derived static characteristics via the Kirchhoff formula in Eq. 8:

$$f(\alpha) = \left(2\sqrt{\frac{C_n}{C_{N\alpha}(\alpha - \alpha_0)}} - 1\right)^2 \tag{8}$$

https://doi.org/10.5194/wes-2025-257 Preprint. Discussion started: 24 November 2025

The resulting  $f(\alpha)$  curve is fitted with a piecewise exponential (Leishman) function, reported in Table 5, yielding the parameters  $\alpha_1$  (f=0.7),  $S_1$  and  $S_2$ . The process is illustrated in Fig. 13 (d). The center of pressure is obtained in a similar fashion from the knowledge of  $C_n$  and  $C_m$ , as in Eq. 9:

$$x_{CP}(\alpha) = \frac{C_m - C_{M0}}{C_n} \tag{9}$$

and subsequently fitted with the semi-empirical formulation of Beddoes (see Table 5). The procedure, as well as the meaning of the different fitting constants  $k_0$ ,  $k_1$ ,  $k_2$ ,  $k_3$  is shown in Fig. 13 (f).

Finally, the static stall threshold  $C_{N1}$ , used in the BL model to mark leading edge separation, is identified as the  $C_n$  value at the maximum  $C_c$  in the pre-stall region. This criterion is deemed to be the most robust among those available in the literature, as the  $C_c$  curve always presents one peak at the static stall point. The latter is also more coherent with the algorithm used in the BL model to reconstruct the  $C_c$  curve (see Table 5), where  $C_{N1}$  is the breakpoint between the formulations used for the pre-and post-stall regions (see Fig. 13 (e)). The shape of the post-stall curve can be further tuned by adjusting the parameters  $D_{CC}$  and  $\hat{k}_1$ , though  $D_{CC}$  is not used in AeroDyn.

#### 4 Annotated comparisons of airfoil families

The operating condition of an airfoil, often expressed in terms of Reynolds and Mach numbers, greatly influence its performance. While this is commonly acknowledged and is not a novelty in itself, the variety of conditions of the examined dataset, which have been tailored specifically to multi-megawatt wind turbine rotors, nevertheless enables the identification of trends directly relevant to rotor aerodynamic design and optimization.

As a first example, lift and drag coefficients, as well as the aerodynamic efficiency, are shown as a function of the angle of attack for the FFA-W3-241 airfoil in Fig. 14. As expected, the Reynolds number has a strong influence on all quantities of interest. For both the lift and the drag coefficients, fully turbulent simulations are more sensitive than transitional ones to the changes in inflow conditions, as general decrease in drag coefficient with increasing Reynolds number can be noted. The stall limit (static stall angle and maximum lift coefficient before stall) increases instead. These changes are reflected in airfoil efficiency, which grows with the Reynolds number in fully turbulent conditions (Fig. 14 (c)).

Figure 14: (a) Lift, (b) drag and (c) aerodynamic efficiency of the FFA-W3-241 airfoil for various Reynolds numbers in fully turbulent and free transition conditions.

More general conclusions can be drawn from Figures 15 and 16, where the design angle of attack, here taken as the point where the airfoil presents the maximum aerodynamic efficiency  $E = C_l/C_d$ , is shown together with the corresponding aerodynamic efficiency and lift coefficient as a function of Reynolds and Mach number. The same trends are reported for the static stall angle. These are considered the key parameters for assessing the performance of an aerodynamic section used in a wind turbine rotor. Lift-to-drag ratio is of paramount importance to minimize the penalty on torque (drag) per unit of useful aerodynamic force (lift). In a horizontal axis wind turbine, this is especially important in the outboard sections of the blade, where, due to the blade's high tangential velocity, lift is directed mostly out of plane, thereby increasing unwanted thrust force rather than the driving torque, and drag is directed mostly in-plane, accentuating its detrimental effect on torque. The maximum airfoil efficiency for airfoils with 30% or lower relative thickness in fully turbulent conditions is shown in Figure 15 as a function of Reynolds number (c) and Mach number (g), respectively. For all airfoil families, lower thickness ratios generally lead to higher aerodynamic efficiency. For all airfoils, efficiency tends to increase with the Reynolds number. However, particularly for more outboard, thinner airfoils, efficiency plateaus with very little, if any, increase beyond 15\*106 Reynolds number. From an inspection of Figure 15 (g), compressibility appears to influence this effect. In fact, while the Reynolds number is the same for all airfoils and ranges from 2.5\*106 for 2\*107, the Mach number is not as thinner airfoils are simulated with higher inflow velocities. If maximum efficiency is shown as a function of the Mach number (Fig. 15 (g)), with the only exception of the DU91-W2-250 and FX77-258 airfoils, which underperform the rest, all airfoils follow a very similar trend. When compressibility effects are low and the efficiency curves remain below Ma=0.2 (as is the case for the FFA-W3-301 airfoil for example), efficiency increases as the Reynolds (and Mach) number increase. Above Ma=0.25 however, the maximum lift to drag ratio starts to plateau for all the tested airfoils, the FFA-W3-211 airfoil being a clear example of this trend. Both the Reynolds number and compressibility have a marked effect on stall performance. In fact, both the stall angle

(Fig. 15 (c)), and the design angle (Fig. 15 (f)) tend to initially increase at lower Reynolds numbers, due to the increased resistance to stall as the airfoil boundary layer becomes increasingly thinner, but tend to decrease at higher Mach numbers, highlighting the effect of compressibility.

Figure 15: (a, e) stall angle computed as angle of attack of maximum lift (b, f) Design angle of attack, (c, g) maximum aerodynamic efficiency, (d, h) lift at design angle of attack. Quantities shown as function of the Reynolds number (a-d) and Mach number (e-h). Fully turbulent simulations of airfoils with less than 30% relative thickness.

The variation in design lift coefficient as a function of the Reynolds and Mach number is also significant, with maximum differences exceeding 20% in some cases. The differences are stronger for more inboard airfoils, which are less affected by flow compressibility. These variations may have a significant impact on design, as blade sections can experience significant variations in Reynolds and Mach number depending on the rotational speed.

The same analysis is repeated in free transition conditions. The stall angle, the design angle of attack, as well as the lift coefficient at the design angle of attack and the maximum airfoil efficiency are shown as a function of the Reynolds and Mach number for various airfoil families in Fig 16. While the effect of compressibility on the stall angle, which generally tends to be anticipated with increasing Mach number, is similar to what noted in fully turbulent conditions, the variations in design angle of attack, maximum efficiency, and design lift coefficient follow different trends. When the boundary layer is allowed to transition, the model predicts greater resistance to stall, as demonstrated by the higher stall angles, especially at lower angles of attack. Resistance to stall is less influenced by the Reynolds number and appears to be more airfoil dependent. In addition,

aerodynamic efficiency follows a different trend as a function of Reynolds number with respect to what was noted in fully turbulent conditions (Fig. 16 (c)) and generally decreases as Reynolds number increases. This trend is also confirmed by experiments, although inflow characteristics, and particularly inflow turbulence, varied slightly in the AVATAR test campaign (Pires et al., 2016). The reason for the decrease in maximum efficiency can be explained by examining Fig. 16, where the pressure (Cp) and skin friction (Cf) coefficients for the FFA-W3-241 airfoil in both free transition and fully turbulent conditions are shown. In both cases, pressure loading in the front region of the airfoil suction side (x/c < 0.4) tends to increase with Reynolds, due to a reduction in boundary layer thickness. In fully turbulent conditions, this increase is met by a corresponding decrease in skin friction, explaining the increased maximum efficiency. When the boundary layer is allowed to transition from laminar to turbulent however, the higher the Reynolds number, the earlier the flow tends to transition, as shown by the increases in skin friction in Fig. 17 (b). The earlier laminar to turbulent transition reduces the low-friction area of the airfoil operating with a laminar boundary layer. The anticipated transition point is balanced by the decrease skin friction in both the laminar and turbulent regimes at higher Reynolds numbers (Fig. 17 (b)), ultimately leading to the trends shown in Fig. 17 (c).</li>

Figure 16: (a) (a, e) stall angle computed as angle of attack of maximum lift (b, f) Design angle of attack, (c, g) maximum aerodynamic efficiency, (d, h) lift at design angle of attack. Quantities shown as function of the Reynolds number (a-d) and Mach number (e-h). Free transition simulations of airfoils with less than 30% relative thickness. Experimental data from (Ozlem et al., 2017) for reference, not computed in the same Reynolds and Mach conditions.

A similar trend in terms of the variation in maximum efficiency and design angle of attack as a function of the Reynolds number is also noted in the experiments performed by (Ozlem et al., 2017) on the DU-00-W-212 airfoil. The experimental

results are compared in Fig. 18 to the same set of airfoils shown in Fig. X. In the experiments performed on the DU-00-W-212 airfoils efficiency peaks at Re=9\*10<sup>6</sup> (Fig. 18 (b)), while the angle of attack at which the maximum lift to drag ratio is located (Fig. 18 (a)) decreases as the Reynolds number increases before starting to increase at Re=1.5\*10<sup>7</sup>. These trends are also noted in the simulations, with some differences from airfoil to airfoil and depending on the inflow conditions. The Mach number is also much higher in the simulations, as it ranges between 0.01 and 0.46, while it does not exceed 0.1 in the experimental reference.

Figure 17: (a) Pressure coefficient and (b) skin friction for the FFA-W3-241 airfoil in fully turbulent conditions (blue) and free transition conditions (red) at an angle of attack of 8°.

Figure 18: Comparison of simulated trends (gray lines) in free transition conditions with experimental results on the DU-00-W-212 airfoil from (Ozlem et al., 2017) (black markers). (a) Design angle of attack and (b) maximum efficiency.

The absolute difference between the fully turbulent and free transition cases in terms of stall and design angle of attack, aerodynamic efficiency, and design lift coefficient are shown in Fig. 19. Small differences between the performance in fully turbulent and in free transition conditions are generally desirable. These two extreme conditions lie at either end of the operational windows of a wind turbine blade, and correspond to a dirty, worn-down blade or a new, clean blade, respectively. The larger the difference in terms of efficiency between these two configurations, the larger the performance degradation during the turbine lifetime will be. As shown in Fig. 19, there are generally larger differences between the free transition and fully turbulent scenarios at lower Reynolds numbers. This trend is most noticeable when examining maximum efficiency, as the gap between the free transition and fully turbulent cases closes as the fully turbulent efficiency increases.

Figure 19: (a, e) stall angle computed as angle of attack of maximum lift (b, f) Design angle of attack, (c, g) maximum aerodynamic efficiency, (d, h) lift at design angle of attack. Quantities shown as function of the Reynolds number (a-d) and Mach number (e-h). Difference between values in free transition conditions and in fully turbulent conditions.

The operating range of some of the airfoils in the database is compared in Fig. 20, by showing the ranges of angles of attack where efficiency exceeds 85% of the maximum value at each Reynolds number. A broad high-efficiency operating range is desirable in wind turbine applications as these machines operate in highly turbulent inflow and face highly variable inflow conditions, which cause the angle of attack to vary during operation. In addition, modern wind turbine blades tend to deform as rotor load increases, due to combined effects of aerodynamic loading and aeroelastic phenomena such as bend-twist and

shear-twist coupling. As such, a broader operating range can increase broaden design options and the optima operational window. When fully turbulent and free transition conditions are compared (top and bottom row of Fig. 20, respectively), the range of angles of attack with good performance remains similar, but the stall margin, i.e., the distance of this optimal operating range from the static stall point, is higher in the case of a clean blade. In free transition conditions, however, some airfoils such as the DU08-W-210, OSO-21, and DU91-W2-250 suffer from reductions in the optimal operating window at high Reynolds number due to the effects of compressibility, which leads to early stall and performance degradation. FFA-W3 airfoils generally show good performance in this regard.

Figure 20: Angles of attack where efficiency exceeds 85% of the maximum value (lines), angle of attack of maximum efficiency (filled markers) and stall angle of attack (cross) for various airfoils. The ranges for the angle of attack are shown for the five Reynolds numbers in the dataset. The y-values correspond to the maximum efficiency. (a, d) 21% thick airfoils, (b, e) 24-25% thick airfoils, (c, f) 30% thick airfoils. (a, b, c) fully turbulent conditions and (d, e, f) free transition conditions.

#### 4 Conclusions

This paper describes a database of wind turbine airfoil aerodynamic coefficients. The aerodynamic coefficients are computed using a state-of-the-art CFD methodology, which has been fine-tuned and validated based on available experimental data. Results include the effects of compressibility, and computations are performed with and without laminar-to-turbulent transition. The total computational resources required to compute the database exceed 100000 CPU-hours on modern hardware (dual AMD-EPYC 7413 server). The quantity and fidelity of the generated data allow for the effects of Reynolds and Mach

https://doi.org/10.5194/wes-2025-257

Preprint. Discussion started: 24 November 2025

© Author(s) 2025. CC BY 4.0 License.

number variation to be investigated across different families of wind turbine airfoils featuring relative thicknesses between 21% and 36%. More in detail, these effects are more pronounced when considering fully turbulent computations rather than in the free transition cases. In fact, when not considering boundary layer transition, peak efficiency increases as the Reynolds number increases, but benefits are greater at lower Mach numbers. Above a free-stream Mach number of 0.2, efficiency gains appear to be limited by the effects of compressibility. In free transition conditions, the variation in the transition point due to the change in inflow conditions can compensate for the effects of Reynolds and Mach number variation, and the variation in airfoil efficiency as a function of these non-dimensional parameters is much more case-dependent. Because of these trends, the difference in efficiency between the free transition case, representative of a clean blade, and the fully turbulent case, representative of a soiled blade, decreases as the Reynolds number increases. Based on this evidence, the benefits of operating a clean blade are more strongly felt at lower wind speeds, where the Reynolds number is lower, rather than at higher wind speeds, where the Reynolds number increases due to the higher rotational speed. Finally, the more recent airfoils included in this database, namely the OSO family, outperform other academic open-access families such as the FFA-W3 and DU airfoils over a large range of relative thicknesses, proving that modern design tools can indeed improve blade sectional performance. On the other hand, the variation in design angle of attack as a function of the Reynolds number and boundary layer modelling (free transition compared to fully turbulent) are more significant than the OSO family of airfoils, especially for the geometries with higher relative thickness. Finally, it is worth remarking that the presented dataset is to be intended as a "living" dataset, and new geometries may be added to it in the future based on the wind energy community's requests.

# Data availability

The entire dataset is available in open access mode at https://doi.org/10.5281/zenodo.15706283

## **Authors contributions**

FP carried out the validation of the numerical set-up and the numerical simulations, prepared the first draft, and post-processed the results. FP and PFM prepared the numerical set-up. FP and PFM performed software development for the automatized post-processing of airfoil polars. PFM and AB provided supervision of numerical analysis. All the authors contributed to the analysis of the results and editing of the manuscript.

## **Competing interests**

At least one of the (co-)authors is a member of the editorial board of Wind Energy Science.

https://doi.org/10.5194/wes-2025-257 Preprint. Discussion started: 24 November 2025 © Author(s) 2025. CC BY 4.0 License.

# Acknowledgements

The authors wish to thank Prof. Roberto Pacciani and Prof. Michele Marconcini of the University of Florence for the insights and discussions regarding natural boundary layer transition and to Stephen Orlando and Valerio Viti from Ansys Inc. for the technical support in the simulations. Thank you also to Ansys Inc. for the scientific partnership with the authors' research group.

# Financial support

This work has received the support of the FLOATFARM Project, funded by the European Union's Horizon Europe research and innovation programme under grant agreement No. 101136091. Views and opinions expressed are however those of the author(s) only and do not necessarily reflect those of the European Union or the European Climate, Infrastructure and Environment Executive Agency (CINEA), which cannot be held responsible for them.

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
