# Peer review of "An Open Database of High-Fidelity, Multi-Reynolds Airfoil Polars for Wind Turbine Blade Design"

_Wind Energy Science, 2025_

## Referee Comment (RC2)

This paper presents a very interesting and sound methodology for constructing a comprehensive database of aerodynamic characteristics for a variety of wind turbine airfoils covering the full angle-of-attack range from –180° to +180°. The authors generate high-fidelity results using CFD for the conventional operating angle of attack range—approximately between the negative and positive stall angles—and subsequently extend these results to the complete ±180° domain through an empirically based extrapolation procedure. The overall approach is robust, clearly explained, and results in an open database that is highly valuable for lifting line calculations of wind turbines which require such airfoil data.

In addition to the remarks provided by my co-reviewer, I would like to offer a few further comments and questions.

- Line 10: Beyond the stall region... This statement becomes clearer if you explain more explicit that the before mentioned CFD airfoil coefficients are generated for the conventional aoa range only, i.e. from negative to positive stall.
- Line 26: Reference *Bussel and W?*
- Figure 1 This is certainly an intriguing figure, but it raised so many questions for me that it became more distracting than informative. It made me wonder whether including it is worth the effort—especially since its overall purpose and message are not entirely clear to me. Apart from the non-linear time axis, which looks unusual—and where, for example, the DU airfoils appear to be placed around 1991 even though some of them, such as the DU-00-212 used in the follow-up DNW-HDG test in your paper, were developed later—my main question is how the numerical values were determined. Several DU airfoils exist with 21% and 18% thickness, so which specific variants were used to generate these numbers?
- Figure 1: Bending instead of bendig
- Line 69: CENER instead of CNER
- Line 76: Between the lines, you refer to 3D rotational effects, which cause the aerodynamic behavior of a rotating blade to differ from the 2D airfoil characteristics as generated by you. Many readers may think: *"It's useful to have these 2D airfoil data, but what is their relevance for actual wind turbine operation?"* It would therefore be helpful to elaborate more explicitly on the value of 2D airfoil data.
    - I would start by identifying the sources of deviation between 2D airfoil performance and the performance on a rotating blade. Beyond 3D rotational effects, there are also unsteady aerodynamic phenomena (your data are time-averaged) and 3D geometric influences, which may be particularly relevant for quantities such as $c_d(90°)$, as discussed below.
    - Nonetheless, as far as I know all methods which correct for 3D rotational effects, geometric effects, or unsteady airfoil behavior still rely on steady 2D airfoil data as their fundamental input. This means your dataset has substantial value (especially in the context of design approaches based on lifting-line models. It may be worthwhile to mention lifting-line models explicitly at an appropriate point in the manuscript).
- A question on Line 95 and further, on the maximum value of cd: Are we comparing apples with apples? I wonder whether perhaps some cd values are 2D (e.g. the values around 1.8) where others may be on a blade with finite aspect ratio which may lead to cd values around 1.3?

- Line 104: you may be a bit sharper in motivating value of post stall data. It is not only for ultimate blade load estimation in parked conditions but these data may be of relevance for understanding stall and vortex induced vibrations as well. Even though these are unsteady effects I think that steady 2D data form the basis for modelling these effects.
- Line 120: You mention that airfoils with t/c > 21% are included only, but the last one in table 1 has a thickness of 18%
- Line 147: You mention the effect of compressibility on the slope of the lift coefficient. Isn't the drag rise equally important (e.g. use the Prandtl Glauert relation for this drag rise)
- Table 3: The abstract says that simulations are done for conditions which are typical for utility scale wind turbines. Still I wonder whether tip speeds of 153.3 m/s are typical for modern wind turbines. This would lead to a significant drag increase from the Prandtl Glauert relation but also to significant erosion issues. It is more than fine to do simulations for a very wide range of conditions but could you explain the reasons why you have chosen these conditions?
- Section 3.3: I remember some issues in the AVATAR project wrt the y-Re theta model at the high Reynolds numbers test, which was the reason why many participants switched to the $e^N$ transition model or they modified the model. It seems you apply some modifications as well but it may be good to explain how your transition modelling relates to the transition modelling in these references:

  *Ceyhan, O., Pires, O., Munduate, X., Sørensen, N., Schaffarczyk, A. P., Reichstein, T., Diakakis, K., Papadakis, G., Daniele, E., Schwarz, M., Lutz, T., & Prieto, R. (2017). Summary of the blind test campaign to predict the high Reynolds number performance of DU00-W-210 airfoil. In **35th AIAA Wind Energy Symposium** (AIAA 2017-0915), Grapevine, TX, USA.*

  *Colonia, S., Leble, V., Steijl, R., & Barakos, G. (2017). Assessment and calibration of the γ-equation transition model for a wide range of Reynolds numbers at low Mach. **AIAA Journal, 55**(4), 1126–1139. https://doi.org/10.2514/1.J055403*

- Line 254: I agree that modern wind-turbine airfoils—as a matter of fact earlier generations of wind-turbine airfoils as well—experience effectively lower turbulence levels because the rotational component contributes to the resultant velocity in the denominator of the turbulence intensity. However, in my view, the turbulence intensity scales approximately as ~ $I_{ambient}/\lambda_r$ For an ambient turbulence level of say 10% and a local tip speed ratio of say 5 the resulting turbulence intensity would be 2%, much lower than 10% indeed but still twice the threshold value of 1% mentioned by you.
- Figure 12: Good to compare with SBES simulations. A question: Have you considered a comparison with measurements? I am not sure if there are measurements for -180+180 at 10M but could a comparison at lower Reynolds have value too? E.g DNW-HDG measurements from -90 to +90 degrees at RE = 6 M or others?
- Line 358: derived from THE static characteristics?
- I suggest adding a brief description of the AVATAR experiment at an appropriate point in the text to provide readers a bit of context. For example, you could mention that the campaign involved detailed pressure-distribution measurements, supplemented by drag data obtained from a wake rake in the pressurized DNW-HDG wind tunnel. Despite the

model having a chord length of only 15 cm, the pressurization enabled testing at a variety of Reynolds numbers up to 15 million where it is noted that the tunnel turbulence was different (or something like that)

- In the abstract, you mention that differences with other open-access datasets are discussed. To me, that discussion feels somewhat limited. You might consider referring to the Aerodynamic Table Generator developed at TNO (formerly ECN):
Bot, E. T. G. *Aerodynamic Table Generator. A Manual.* ECN-C-01-130, December 2001 (in Dutch unfortunately). Aerodynamische Tabel Generator This tool produces aerodynamic characteristics for a wide range of airfoils based on wind-tunnel measurements for the conventional angle-of-attack range and then extrapolates to −180° to +180° using an empirical method. In that sense, its approach is quite similar to yours. The key difference, of course, is that it relies directly on measurements—an advantage in terms of realism, but also a limitation because such measurements exist for only a restricted set of conditions, resulting in far less flexibility than your method.

Good luck withing finalizing this excellent work!